# COX7A1-mediated mitochondrial dysfunction can induce ferroptosis in endometrial cancer cells

Qi Wu●*, Suning Bai, Liyun Song, Lina Han, Pei Wang

Department of Gynecology, Hebei general hospital, Shijiazhuang, Hebei Province, China

* wuqi15511888643@163.com

## Abstract

In endometrial cancer, research on ferroptosis is still in its nascent stages, yet its potential therapeutic value is becoming increasingly evident. We explore the impact of COX7A1 on mitochondrial dysfunction and ferroptosis in endometrial cancer. In this study, through comprehensive bioinformatics analysis, differentially expressed genes related to ferroptosis in endometrial cancer were identified. In vitro experiments were conducted using cytochrome c oxidase subunit 7A1 (COX7A1) overexpression and knockdown cell lines, followed by ferroptosis-related phenotypic assays to validate the effect of COX7A1 on the inhibition of endometrial cancer cell growth. Mechanistically, mitochondrial function-related parameters were assessed to explore the potential mechanisms by which COX7A1 induces ferroptosis. Online data analysis revealed that COX7A1 acts as a ferroptosis driver and is significantly downregulated in endometrial cancer tissues. In vitro experiments have demonstrated that overexpression of COX7A1 inhibits the proliferation of endometrial cancer cells and induces ferroptosis by regulating intracellular iron metabolism and mitochondrial function. The specific mechanisms include increasing intracellular $Fe^{2+}$ and malondialdehyde (MDA) levels, decreasing the GSH/GSSG ratio, and disrupting mitochondrial membrane potential, thereby leading to mitochondrial dysfunction. Furthermore, COX7A1 overexpression significantly reduces the expression of glutathione peroxidase 4 (GPX4) and SLC7A11, while upregulating acyl-coenzyme A synthetase long-chain family member 4 (ACSL4). In contrast, knockdown of COX7A1 promotes the proliferation of endometrial cancer cells and inhibits ferroptosis, exhibiting the opposite effects. These findings provide new insights into the molecular mechanisms of endometrial cancer.

## Introduction

Endometrial cancer is one of the most common malignant tumors in the female reproductive system, with a rising incidence rate observed globally, especially in developed countries [1]. According to recent statistics, endometrial cancer ranks sixth among all

**Data availability statement:** The datasets generated and/or analyzed during the current study are available in the Figshare repository at: https://figshare.com/s/5678c40e643bf55f363e.

**Funding:** This study was funded by the Medical Science Research Project of Hebei Province (grant no. 20211491), received by Dr. Qi Wu.

**Competing interests:** The authors have declared that no competing interests exist.

malignancies in women worldwide, and its mortality rate has been increasing annually [2]. Approximately 400,000 new cases are diagnosed each year [3], resulting in around 76,000 deaths [4]. Although advancements in early diagnosis and surgical techniques have improved patient survival rates, the prognosis for patients with advanced or recurrent disease remains poor [5]. Currently, the primary treatments for endometrial cancer include surgery, radiotherapy, and hormone therapy [6]. However, these therapies often have limited efficacy in managing advanced or recurrent tumors and are prone to drug resistance [7]. Therefore, developing new treatment strategies to improve the prognosis of patients with advanced and recurrent endometrial cancer is a critical focus in current research.

In recent years, ferroptosis, a new form of programmed cell death characterized by iron-mediated lipid peroxidation, has garnered significant attention. It is intricately linked with the onset, progression, and therapeutic strategies for tumors [8,9]. Studies have shown that inducing ferroptosis may offer a promising alternative for treating endometrial cancer, particularly when traditional therapies prove ineffective [10]. By regulating intracellular iron metabolism and antioxidant systems, ferroptosis triggers the accumulation of reactive oxygen species (ROS), thereby causing cell death [11]. This mechanism presents innovative approaches for managing endometrial cancer, especially in addressing drug resistance and recurrence. Although research on ferroptosis in endometrial cancer is still nascent, its potential therapeutic value has gradually emerged. Existing studies have demonstrated that endometrial cancer cells show different degrees of sensitivity to ferroptosis inducers such as sodium butyrate and PTPN18, indicating the potential of ferroptosis as a therapeutic target [12,13]. However, numerous challenges remain. The precise regulatory mechanisms of ferroptosis, especially the specific regulatory network within endometrial cancer, have yet to be fully elucidated [14,15]. Moreover, mitochondrial dysfunction, which plays a pivotal role in ferroptosis, warrants further investigation. Mitochondria, serving as the cellular energy hub, also regulate oxidative stress and lipid metabolism, processes that are intimately associated with ferroptosis [16,17]. An in-depth study of mitochondrial dysfunction's role in endometrial cancer is crucial for unraveling the regulatory mechanisms of ferroptosis.

To thoroughly explore the role of ferroptosis-related genes in endometrial cancer, the GEO database was initially utilized to identify differentially expressed genes (DEGs) related to ferroptosis. This analysis led to the identification of nine intersection genes, among which COX7A1 caught our attention. COX7A1 is a subunit of mitochondrial respiratory chain complex IV. Its dysfunction can lead to mitochondrial energy metabolism disorders, resulting in ROS accumulation and subsequent cellular ferroptosis [18]. Our findings revealed that COX7A1 expression was significantly downregulated in endometrial cancer tissues and strongly correlated with the expression levels of other ferroptosis-related genes. To further elucidate the functional significance of COX7A1, in vitro experiments were conducted using endometrial cancer cell models with COX7A1 knockdown and overexpression. Various parameters, such as cell viability, lipid peroxidation levels, mitochondrial membrane potential, and ROS levels, were measured. These experiments aimed to investigate the relationship

between COX7A1-mediated mitochondrial dysfunction and ferroptosis. This research is of great significance for revealing the pathogenesis of endometrial cancer and developing novel therapeutic strategies.

## Methods and materials

### Data acquisition and differential gene analysis

Three transcriptome datasets pertaining to endometrial cancer were retrieved from the Gene Expression Omnibus (GEO) database (https://www.ncbi.nlm.nih.gov/gds): GSE63678 (comprising 5 normal and 7 tumor cases), GSE106191 (including 33 normal and 28 tumor cases), and GSE115810 (encompassing 3 normal and 24 tumor cases). To eliminate batch effects, the "sva" package in R software (version 4.2.1) was used, and Principal Component Analysis (PCA) plots were generated both before and after batch effect removal [19]. Differential expression analysis between the normal and tumor groups was conducted using the "limma" package in R software [20], with filtering criteria set at |logFC| > 0.585 and FDR < 0.05. The top 30 differentially expressed genes were visualized via volcano plots and heatmaps using the "ggplot2" package [21].

### Screening of potential ferroptosis genes

The ferroptosis-related gene set was retrieved from the FerrDb database (http://www.zhounan.org/ferrdb/current/). A Venn diagram, generated using the VennDiagram package, illustrated the intersection between Differentially Expressed Genes (DEGs) and ferroptosis-related genes [22]. Additionally, a Sankey diagram, created with the "ggsankey" package in R, visualized the functional classification of these intersection genes as either drivers or inhibitors of ferroptosis [23]. The differential expression profiles of the nine intersection genes in cancer and adjacent tissues were obtained from the TCGA-UCEC cohort (https://portal.gdc.cancer.gov/).

### Cell culture

Human endometrial epithelial immortalized HEEpiC-SV40 cells were purchased from Stemmer Biotech Co., Ltd. Human endometrial cancer cells Ishikawa, RL95–2, AN3 CA, and KLE were sourced from Xiamen Immocell Biotechnology Co., Ltd., while HEC-50B was acquired from Qingqi Shanghai Biotechnology Development Co., Ltd. Cells were rapidly thawed by transferring them from liquid nitrogen to a 37°C water bath (110805002, BKMAMLAB, China) for 5 minutes. They were subsequently cultured in Dulbecco's Modified Eagle Medium (DMEM) medium (C3117-0500, VivaCell, Israel) supplemented with 10% fetal bovine serum (F2442, Sigma, Germany) and 1% (v/v) penicillin-streptomycin (CPS101.02, CellMax, China). The cultures were maintained in a humidified incubator at 37°C with 5% $CO_2$ (BX-100CB, Shanghai Boxun Medical Biological Instrunent Corp., China). Cells in the exponential growth phase were utilized for subsequent experiments.

### Construction of COX7A1 knockdown and overexpression cell models

Ishikawa and AN3 CA cells in the exponential growth phase were harvested. The culture medium was aspirated, and cells were washed with phosphate-buffered saline (PBS). A 0.25% trypsin-EDTA solution (40101ES25, Yeasen Biotechnology (Shanghai) Co., Ltd., China) was added for a 3-minute incubation. After discarding the trypsin solution, DMEM medium was added to resuspend the cells. Cell density was determined using a cell counter (CytScop® Mini, BioAces (Shanghai) Life Science Co., Ltd., China), and cells were seeded at a density of $1 \times 10^5$ per well in a six-well plate. According to the manufacturer's protocol for Lipofectamine 3000 transfection reagent (L3000001, Thermo Fisher Scientific, USA), the COX7A1 overexpression plasmid was transfected into Ishikawa cells, while siRNA was transfected into AN3 CA cells [24]. The efficiency of knockdown and overexpression was verified by RT-qPCR and Western blot analysis. Experimental groups included: For Ishikawa cells, oeNC (Overexpression Empty Vector Negative Control) and oeCOX7A1

(overexpression vector group); for AN3 CA cells, siNC (siRNA Negative Control), siCOX7A1-1 (transfected with siRNA-1), and siCOX7A1-2 (transfected with siRNA-2). The overexpression plasmid (FH1750, Hunan Fenghui Biotechnology Co., Ltd., China) and SiRNA sequences (SC2020030900, Hunan Fenghui Biotechnology Co., Ltd., China) were used.

## Cell counting Kit-8

Ishikawa and AN3 CA cells in the exponential growth phase were seeded in a 96-well plate at a density of 3000 cells per well. Cells were treated according to the experimental design at 0, 24, 48, and 72 hours, respectively. Following treatment, 10 µL of CCK-8 working solution (CK001, LABLEAD, China) was added to each well, and the plate was incubated for an additional 4 hours in a cell culture incubator. After the incubation, the plate was transferred to a microplate reader (Sunrise, Tecan, Switzerland) to measure the OD value at 450 nm for further data processing and analysis. For each group in the experiment, subtract the OD value of the well containing only the culture medium and the CCK8 reagent but no cells (blank control) from the OD value of the experimental group. This is to eliminate the influence of background absorption. An increase in the OD value generally indicates an increase in the number of active, metabolically vigorous cells.

## EDU (5-Ethynyl-2'-deoxyuridine) staining

Ishikawa and AN3 CA cells in the exponential growth phase were harvested. Cell counting was performed as previously described. Cells were seeded at a density of $1 \times 10^5$ per well in a six-well plate and treated according to the experimental design for each group. After adding the EDU working solution (C10310-1, RiboBio, China), the cells were placed in the cell incubator for further incubation for 2 hour. Following incubation, the EdU solution was removed, and cells were fixed with 4% paraformaldehyde fixative for 30 minutes at room temperature. After discarding the fixative, 4',6-diamidino-2-phenylindole (DAPI) staining solution (C1002, Beyotime, China) was added, and the cells were incubated for an additional 5 minutes. The cells were then washed three times with PBS. Fluorescence images were captured using an inverted fluorescence microscope (Leica DMi8, Leica Microsystems, Germany) for subsequent statistical analysis. Fluorescent images were captured using an inverted fluorescence microscope (Leica DMi8, Leica Microsystems, Germany). For each group, 5 visual fields were randomly selected, and the proportion of EDU-positive cells in the images was statistically analyzed using Image J 1.5.2a software.

## JC-1 fluorescence

Ishikawa and AN3 CA cells in the exponential growth phase were collected. Cell counting followed previously established methods. Cells were seeded at a density of $2 \times 10^5$ cells per well in a six-well plate and subjected to treatments as specified for each experimental group. Following the manufacturer's instructions for the JC-1 staining kit (E-CK-A301, Elabscience, China), the JC-1 working solution was added, and the cells were incubated for 30 minutes in a cell culture incubator. The stained cells were examined using an inverted fluorescence microscope, and images were obtained for subsequent statistical analysis. For flow cytometry analysis, the JC-1 staining procedure was identical to the aforementioned method. After the incubation, the cells were transferred to flow cytometry tubes and analyzed via flow cytometry within a 2-hour timeframe. The proportions of red fluorescence and green fluorescence in the cells were analyzed using Image J 1.5.2a, and the ratio of red fluorescence to green fluorescence was calculated.

## ROS fluorescence

Ishikawa and AN3 CA cells were seeded in a six-well plate at a density of $1 \times 10^5$ cells per well. Cells were treated as specified for each experimental group. The culture medium was replaced with 10 µmol/L 2',7'-dichlorodihydrofluorescein diacetate (DCFH-DA) working solution (MX4802−50MG, Shanghai Maokang Biotechnology Co., Ltd., China), and the cells were incubated for 30 minutes. Following incubation, cells were trypsinized for 3 minutes. The resulting cell suspension was collected in a centrifuge tube and centrifuged at 1200 g for 3 minutes. After discarding the supernatant, cells

were resuspended in PBS and transferred to flow cytometry tubes for analysis within 2 hours using a flow cytometer. Data on ROS detected by flow cytometry were systematically analyzed using Flow Jo v10.8.1.

## Determination of MDA (Malondialdehyde), $Fe^{2+}$ and GSH/GSSG (Reduced glutathione/oxidized glutathione)

Cells were seeded in a six-well plate at a density of $2 \times 10^5$ cells per well and treated according to the specific requirements of each experimental group. After treatment, cells were trypsinized and centrifuged at 1200 g for 3 minutes to obtain cell pellets. The old medium was discarded. The MDA (G0109W, Suzhou Grace Biotechnology Co., Ltd., China) and $Fe^{2+}$ working solutions were prepared according to the kit instructions. For MDA detection, the working solution was added, and the cells were transferred to a centrifuge tube and incubated in a water bath at 100°C for 20 minutes. Following incubation, the samples were cooled to 25°C under running water. A volume of 0.2 mL of the supernatant was transferred to a 96-well plate and analyzed using a microplate reader (WD-9417B, Beijing Liuyi Biotechnology Co., Ltd., China) to measure the OD value at 532 nm. For $Fe^{2+}$ detection, the working solution was added, and the plate was incubated at 37°C in the dark for 1 hour before measuring the OD value at 593 nm. To determine GSH/GSSG content, the assay was performed according to the instructions of the GSH/GSSG kit (A0601-1, Nanjing Jiancheng Bioengineering Institute, China). After adding the prepared working solution, the OD value was read at 405 nm using the microplate reader. All data obtained from the microplate reader were saved for subsequent statistical analysis.

## Electron microscopy observation

Following the treatment of each cell group with the above methods, the cells were sequentially fixed in 2.5% glutaraldehyde fixative solution (R27496, Shanghai Yuanye Bio-Technology Co., Ltd., China) and 1% osmic acid fixative solution at 4°C for 2 hours each. Subsequently, the samples underwent dehydration through a graded ethanol series. The ethanol was then replaced with acetone, followed by embedding in epoxy resin at a 1:1 ratio, which was allowed to stand at 25°C for 2 hours. Ultrathin sections of 50 nm were prepared using an ultramicrotome (UM10, Lebo Science, China). These sections were stained with uranyl acetate and lead citrate (L26640-5g, Shanghai Acmec Biochemical Technology Co., Ltd., China) at 25°C for 5 minutes each, rinsed with distilled water, and air-dried. The morphological changes in mitochondria were examined using a transmission electron microscope (Spectra Ultra S/TEM, Thermo Fisher Scientific, USA), and images were captured.

## RT-qPCR (reverse transcription quantitative polymerase chain reaction) experiment

Cells from different groups were treated as previously described, and cell pellets were obtained through enzymatic digestion followed by centrifugation. Total RNA was extracted using the Cell Total RNA Extraction Kit (G3013-100ML, Servicebio, China) according to the manufacturer's instructions after adding Trizol reagent. The RNA concentration was quantified using a Nanodrop ultraviolet spectrophotometer (NanoDrop2000/2000c, Thermo Fisher Scientific, USA). First-strand cDNA synthesis was performed using the cDNA Synthesis Kit (KR118, TIANGEN, China) under the following PCR conditions: 37°C for 15 minutes and 85°C for 5 seconds. Gene-specific qPCR primers were designed, and real-time fluorescence quantitative PCR was conducted using the Real-Time PCR Kit (FP205−02, TIANGEN, China) on a PCR instrument (4376600, Thermo Fisher Scientific, USA) with the following cycling parameters: initial denaturation at 95°C for 30 seconds, followed by 40 cycles of 95°C for 5 seconds and 60°C for 30 seconds. After the reaction, the Ct values of each sample were recorded, and the relative expression levels of the target genes were calculated using the 2^-ΔΔCt method [25]. Detailed information regarding the qPCR primers is provided in S1 Table.

## Western blot experiment

Cells from different groups were subjected to treatments as previously described, and cell pellets were collected via enzymatic digestion followed by centrifugation. The cells were lysed using radioimmunoprecipitation assay (RIPA) buffer (P0013B, Beyotime Biotechnology, China) on ice for 40 minutes, with intermittent vortexing (SWV-4500B, Servicebio,

China) for 10 seconds every 10 minutes. Following lysis, the samples were heated in a metal bath at 95°C for 15 minutes to denature proteins. The protein concentrations were quantified using a bicinchoninic acid (BCA) protein quantification kit (P0011, Beyotime Biotechnology, China). Equal amounts of protein were adjusted by adding appropriate volumes of RIPA buffer and loading buffer to ensure consistency across all groups. Electrophoresis was performed at 120 V for 90 minutes, loading 25 μg of protein per lane. Proteins were transferred onto polyvinylidene difluoride (PVDF) membranes at a constant current of 260 mA for 60 minutes. Membranes were blocked with 5% skimmed milk powder for 2 hours, washed three times with TBST, and incubated overnight with primary antibodies. Secondary antibodies were added for incubation in the dark for 2 hours, followed by three washes with Tris-buffered saline with 0.1% Tween-20 (TBST). Chemiluminescent detection was conducted after a 10-second incubation with chemiluminescent substrate. Images were captured using a gel imager for subsequent analysis with Image J 1.5.2a software. The antibody information and dilution ratios are detailed in S2 Table.

### Statistical analysis

In this study, statistical analysis was performed using GraphPad Prism 9.5.0. Continuous variables were presented as mean ± standard deviation (Mean ± SD) after being confirmed to follow a normal distribution by the Shapiro-Wilk test. All experiments were completed with at least three independent biological replicates. For comparisons between two groups, an independent-samples Student's t-test (two-tailed) was used. When multiple-group comparisons were involved, one-way ANOVA combined with the Brown-Forsythe test was first used to verify the homogeneity of variance. For data that met the sphericity test assumption, Tukey's honestly significant difference (HSD) multiple-comparison method was used for post-hoc analysis. The statistical significance threshold was set at $\alpha = 0.05$, and all test results reported the exact P-value and 95% confidence interval.

## Results

### Mining potential therapeutic genes for endometrial cancer based on online databases

The endometrial cancer datasets of GSE106191, GSE115810, and GSE63678 were retrieved. Principal component analysis (PCA) revealed that each dataset exhibited clear separation after batch correction (Fig 1A and 1B). A total of 261 DEGs were identified, comprising 191 upregulated and 70 downregulated genes. These findings were visualized using a volcano plot (Fig 1C). The top 30 significantly DEGs were further illustrated via a heatmap. Notably, MMP12, LCN2, and CCL20 were found to be significantly downregulated in cancer, whereas SFRP4, OSR2, and HAND2-AS1 showed significant upregulation (Fig 1D). Subsequently, we obtained 407 ferroptosis-related genes from the FerrDb dataset and intersected them with the DEGs identified in endometrial cancer, as depicted in a Venn diagram. This intersection yielded a total of 9 candidate genes, namely CDO1, COX7A1, DDIT4, DDR2 [26], IDO1, LCN2, NR2F2, RPM2, and SLC6A14 (Fig 1E). Among these, CDO1, COX7A1, DDIT4, and DDR2 were classified as ferroptosis drivers, while IDO1, LCN2, NR2F2, RPM2, and SLC6A14 were categorized as ferroptosis inhibitors (Fig 1F).

We also examined the differential expression of nine candidate genes in endometrial cancer and adjacent tissues. Analysis of the TCGA-UCEC cohort revealed no significant differences in DDIT4 and SCL6A14 between cancer and adjacent tissues, leading to their exclusion from further investigation. The remaining factors exhibited significant differential expression between endometrial cancer and adjacent tissues, making them our primary research focus (Fig 2A). COX7A1, a subunit of the cytochrome c oxidase complex, plays a crucial role in the function of the mitochondrial electron transport chain [18]. Despite limited research on its involvement in ferroptosis and cancer-related diseases, COX7A1's potential role warrants exploration. Although genes such as CDO1, DDR2, IDO1, LCN2, NR2F2, and RRM2 have been extensively studied in the context of ferroptosis and cancer, the specific contributions of COX7A1 remain underexplored. Given its key role in energy metabolism and oxidative stress, COX7A1 may possess unique functions in regulating ferroptosis and

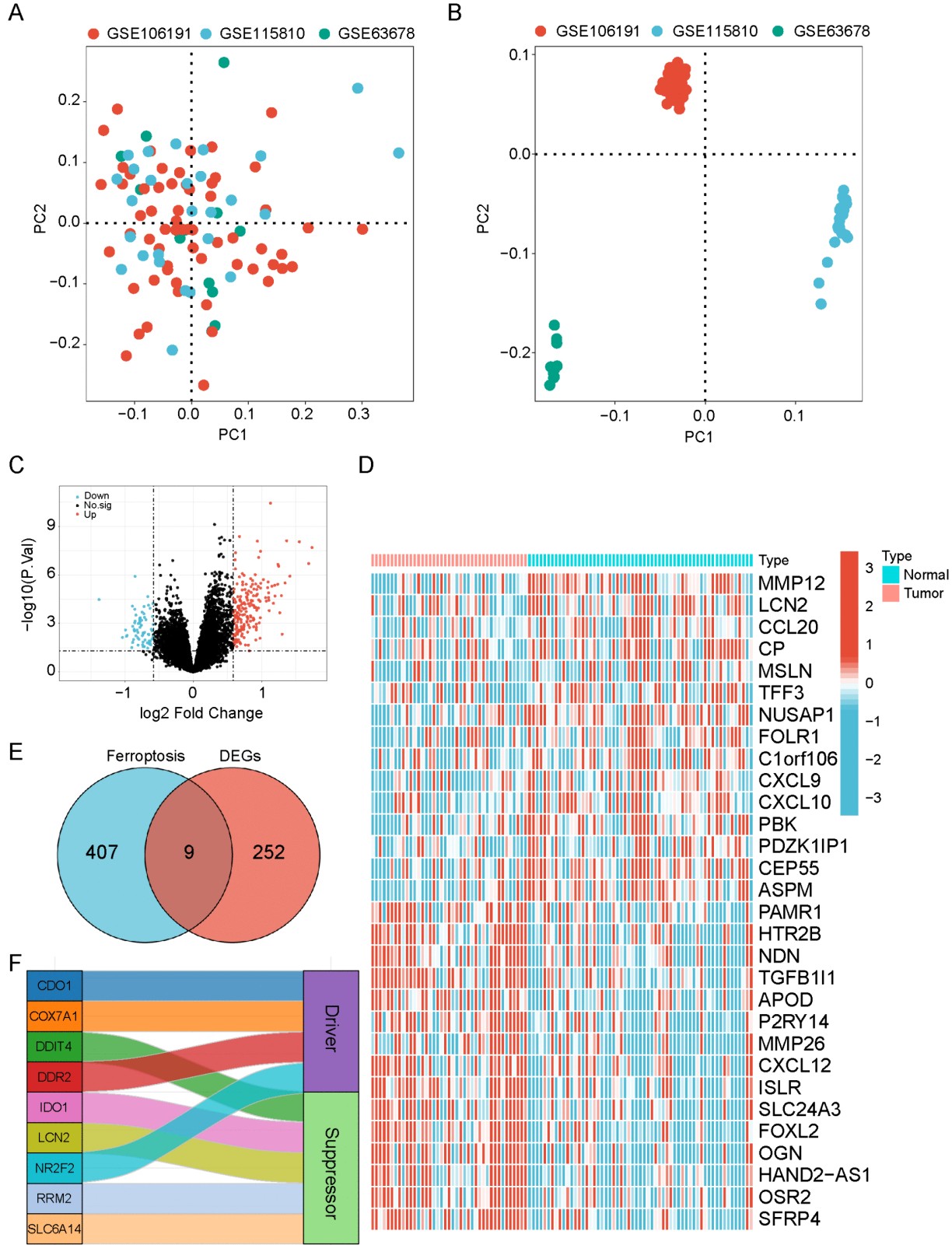

**Fig 1. Screening of candidate therapeutic genes for endometrial cancer.** Principal component analysis (PCA) of the distribution of each dataset before batch removal through the "sva" package. The red dots represent the GSE106191 dataset, the cyan dots represent the GSE115810 dataset, and

the green dots represent the GSE63678 dataset. **(B)** Differentially expressed genes (DEGs) in the datasets are displayed through volcano plots. **(C)** Heatmap display of DEGs. **(D)** The intersection of endometrial cancer datasets and ferroptosis-related datasets is obtained through a Venn diagram. **(E)** Sankey diagram shows the classification of intersection genes (driving ferroptosis or inhibiting ferroptosis).

cancer progression. Therefore, choosing COX7A1 as the focus for subsequent research not only helps to fill the research gap in this field but also provides new insights into the molecular mechanisms of ferroptosis and cancer, which is of great scientific significance and holds potential application value.

## COX7A1 overexpression inhibits the proliferation of endometrial cancer cells

To verify the specific expression profile of COX7A1 in endometrial cancer, a comparative analysis was conducted using five commonly utilized endometrial cancer cell lines and normal cell lines. Our findings revealed that COX7A1 expression was generally lower in endometrial cancer cells compared to HEEpiC-SV40 cells. Notably, the expression of COX7A1 was minimal in Ishikawa cells, while it was relatively higher in AN3 CA cells relative to other endometrial cancer cell lines (Fig 3A). To elucidate the relationship between the expression levels of COX7A1 and the progression of ferroptosis in endometrial cancer, we constructed an overexpression model in Ishikawa cells, which exhibit low expression of COX7A1, and a knockdown model in AN3 CA cells, characterized by high COX7A1 expression. The qPCR and Western blot results confirmed the successful establishment of both the COX7A1 overexpression and knockdown models, and the results were consistent (Fig 3B, 3C).

The EDU assay results showed that compared to the control group, COX7A1 overexpression led to a significant decrease in the proportion of EDU-positive cells, thus inhibiting the proliferation of Ishikawa cells. In contrast, COX7A1 knockdown significantly promoted the proliferation of AN3 CA cells (Fig 4A). These findings suggest that COX7A1 has an inhibitory effect on endometrial cell proliferation. The CCK-8 assay results further corroborated this conclusion, revealing that as incubation time increased, the inhibitory effect of COX7A1 overexpression on Ishikawa cells progressively intensified in a time-dependent manner. Conversely, COX7A1 knockdown markedly enhanced the proliferative ability of AN3 CA cells (Fig 4B).

## COX7A1 induces ferroptosis in endometrial cancer cells

To further investigate the potential link between COX7A1 inhibition of endometrial cancer cell proliferation and ferroptosis, we examined changes in the levels of $Fe^{2+}$, MDA, and GSH/GSSG in endometrial cancer cells. $Fe^{2+}$ is a critical inducer of ferroptosis, while MDA, a product of lipid peroxidation, serves as a biomarker for ferroptosis induction. The GSH/GSSG ratio reflects cellular antioxidant capacity; a decrease in GSH and an increase in GSSG can compromise the cells' resistance to ferroptosis [27,28]. In Ishikawa cells, COX7A1 overexpression significantly elevated intracellular $Fe^{2+}$ levels and increased MDA levels, indicating enhanced lipid peroxidation. In addition, GSH levels decreased while GSSG levels increased, resulting in a reduced GSH/GSSG ratio, which suggests diminished antioxidant capacity. Conversely, in AN3 CA cells, COX7A1 knockdown led to decreased $Fe^{2+}$ and MDA levels, and a restored GSH/GSSG ratio, indicating enhanced cellular resistance to ferroptosis (Fig 5A-C). These findings indicate that COX7A1 can induce ferroptosis in endometrial cancer cells by regulating intracellular iron ion levels and antioxidant metabolism.

We quantified intracellular ROS levels via flow cytometry to assess the induction of ferroptosis in order to further substantiate that COX7A1 can induce ferroptosis. During ferroptosis, excessive accumulation of ROS is a key factor triggering cell death; thus, changes in ROS levels can reflect alterations in intracellular oxidative stress. Additionally, transmission electron microscopy (TEM) was employed to examine mitochondrial structural changes. Ferroptosis can lead to mitochondrial membrane shrinkage and the reduction or disappearance of cristae. These characteristic changes are typical ultrastructural manifestations of ferroptosis [29]. Simultaneously, we assessed the relative expression levels of GPX4, ACSL4, and SLC7A11 proteins. GPX4 is a key antioxidant enzyme that inhibits lipid peroxidation by reducing lipid hydroperoxides,

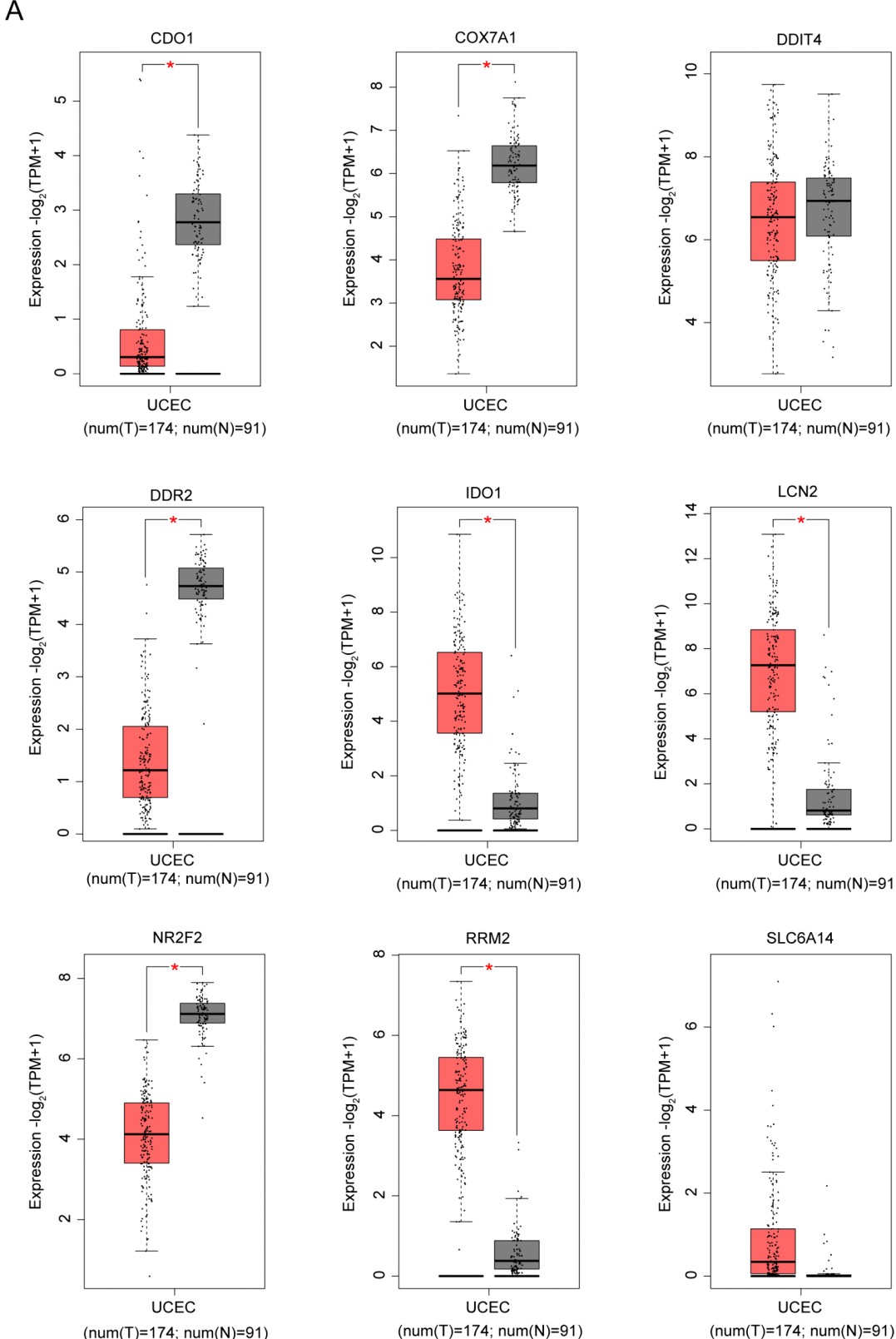

**Fig 2. Differential expression analysis of candidate genes in TCGA-UCEC.** In the TCGA-UCEC (https://portal.gdc.cancer.gov/) cohort, the differential expressions of a total of 9 intersection genes, namely CDO1, COX7A1, DDIT4, DDR2, IDO1, LCN2, NR2F2, RRM2, and SLC6A14, in cancer and

adjacent tissues were obtained. Statistical comparisons between cancer and adjacent tissues are represented (*$P < 0.05$). CDO1: Cysteine dioxygenase type 1; COX7A1: Cytochrome c oxidase subunit 7A1; DDIT4: DNA damage inducible transcript 4; DDR2: Discoidin domain receptor tyrosine kinase 2; IDO1: Indoleamine 2,3-dioxygenase 1; LCN2: Lipocalin 2; NR2F2: Nuclear receptor subfamily 2 group F member 2; RRM2: Ribonucleotide reductase regulatory subunit M2; SLC6A14: Solute carrier family 6 member 14.

thereby protecting cells from ferroptosis damage. ACSL4 enhances cellular sensitivity to ferroptosis by promoting lipid peroxide formation. SLC7A11, an essential component of system Xc⁻, facilitates cystine uptake, affecting glutathione (GSH) synthesis and cellular antioxidant capacity [30]. In Ishikawa cells, COX7A1 overexpression significantly elevated intracellular ROS levels. The mitochondrial ultrastructure in these overexpressing cells exhibited marked membrane shrinkage and reduced cristae, which were highly consistent with the characteristic features of ferroptosis. Western blot analysis showed that COX7A1 overexpression led to a significant reduction in glutathione peroxidase 4 (GPX4) protein expression, a significant increase in acyl-CoA synthetase long-chain family member 4 (ACSL4) expression, and a decrease in the expression of the key subunit SLC7A11 of system Xc⁻. Conversely, COX7A1 knockdown in AN3 CA cells resulted in opposite effects, further confirming that COX7A1 can induce ferroptosis in endometrial cancer cells (Fig 6A-C). These findings suggest that COX7A1 overexpression may promote ferroptosis by downregulating GPX4 and SLC7A11 expression while upregulating ACSL4 expression.

## COX7A1 overexpression induces mitochondrial dysfunction in endometrial cancer cells

To further elucidate the mechanism by which COX7A1 induces ferroptosis, we employed JC-1 probe fluorescence to measure mitochondrial membrane potential (ΔΨm). A reduction in mitochondrial membrane potential is indicative of mitochondrial dysfunction [31]. We also assessed the protein expression levels of VDAC1, Cytochrome C, ATP5A1, and OPA1. VDAC1 (voltage-dependent anion channel 1) is an important protein in the mitochondrial outer membrane that plays a role in regulating apoptosis and ferroptosis; Cytochrome C is an essential component of the mitochondrial electron transport chain, and its release is associated with ferroptosis; ATP5A1 is a subunit of ATP synthase, and its expression level reflects the mitochondrial energy metabolism; OPA1 is a key regulatory protein involved in mitochondrial inner membrane fusion, and changes in its expression can indicate alterations in mitochondrial dynamics [32,33]. Evaluating these factors provides a comprehensive understanding of mitochondrial function changes during COX7A1-induced ferroptosis. After establishing a COX7A1-overexpressing Ishikawa cell line, JC-1 probe fluorescence detection showed a significant decrease in mitochondrial membrane potential, indicating that COX7A1 overexpression leads to mitochondrial dysfunction. Western blot results revealed that the expression levels of VDAC1 and Cytochrome C were markedly elevated, indicative of enhanced permeability of the mitochondrial outer membrane and activation of cell death signals. Concurrently, the expression level of ATP5A1 was significantly decreased, indicating impaired mitochondrial energy metabolism. Additionally, a significant decrease in OPA1 expression was observed, pointing to compromised mitochondrial fusion function. COX7A1 knockdown resulted in opposing effects on mitochondrial membrane potential and the expression levels of associated proteins compared to COX7A1 overexpression (Fig 7A and 7B). Collectively, these findings indicate that COX7A1 overexpression induces ferroptosis through disruption of mitochondrial membrane potential, increased outer membrane permeability, and promotion of ferroptosis.

## Discussion

The genetic mutation profiles and molecular characteristics of endometrial cancer patients exhibit significant heterogeneity, posing substantial challenges for the precise implementation of individualized treatment strategies. Additionally, the lack of reliable prognostic evaluation indicators complicates accurate prognosis prediction and the development of effective treatment plans [34]. To overcome these predicaments, targeting ferroptosis-related genes may offer new insights for the treatment of endometrial cancer. However, given the numerous and functionally complex ferroptosis-related genes,

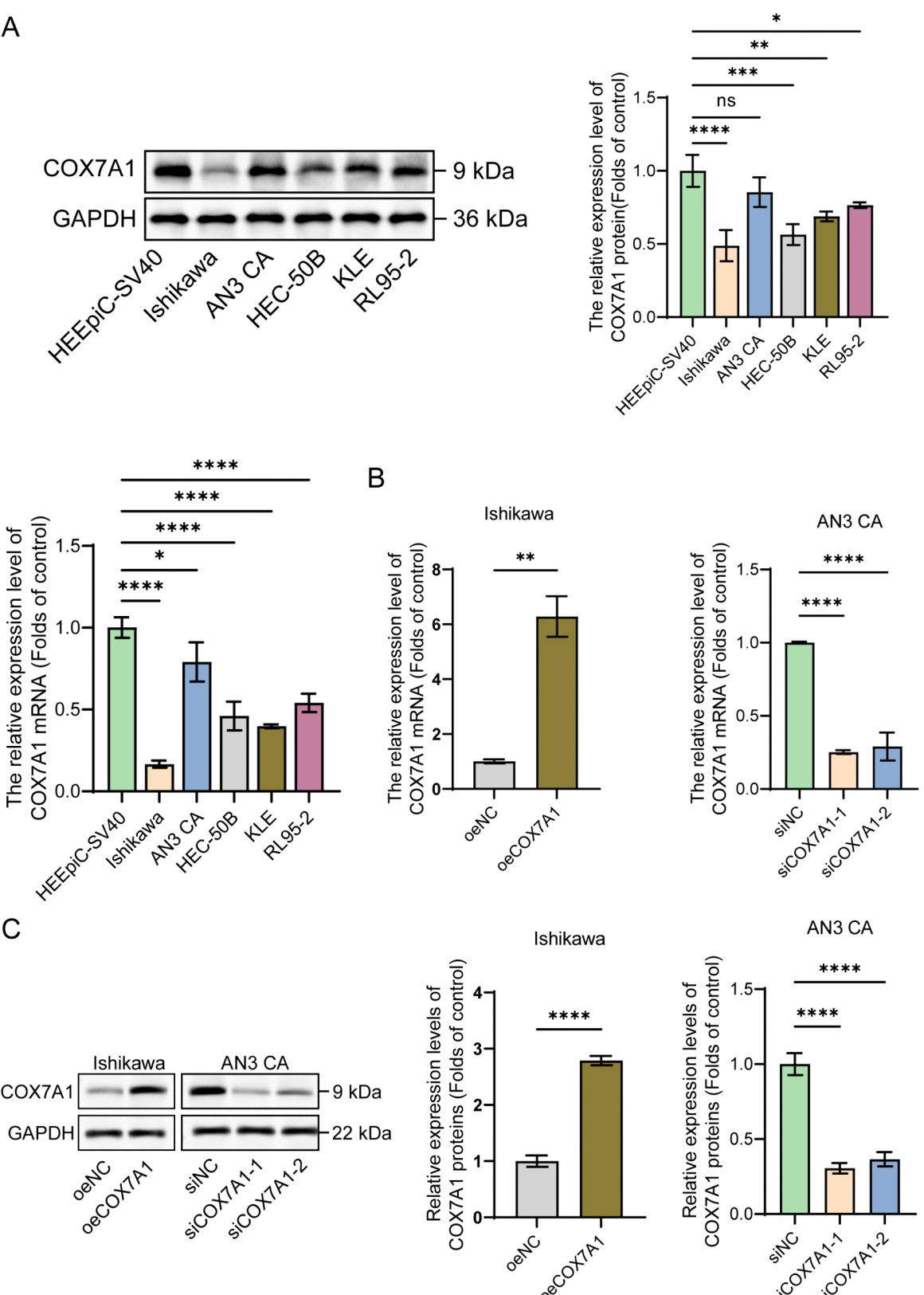

**Fig 3. Low expression of COX7A1 in endometrial cancer.** The relative levels of COX7A1 protein and mRNA were assessed in HEEpiC-SV40, Ishikawa, AN3 CA, HEC-50B, KLE, and RL95−2 cells using Western blot and RT-qPCR experiments (n = 3). **(B)** After transfection with a COX7A1

overexpression plasmid in Ishikawa cells and incubation for 48 hours, as well as transfection with siRNA in AN3 CA cells and incubation for 48 hours, the relative changes in COX7A1 mRNA expression were evaluated by RT-qPCR (n = 3). **(C)** After transfection with a COX7A1 overexpression plasmid in Ishikawa cells and incubation for 48 hours, as well as transfection with siRNA in AN3 CA cells and incubation for 48 hours, the relative changes in COX7A1 protein expression were evaluated by Western blot analysis (n = 3). Statistical comparisons between two groups are indicated. *$P < 0.05$, **$P < 0.01$, ***$P < 0.001$, ****$P < 0.0001$; ns denotes $P > 0.05$. COX7A1: Cytochrome c oxidase subunit 7A1; siNC: siRNA Negative Control; siCOX7A1-1: COX7A1-specific siRNA sequence 1; siCOX7A1-2: COX7A1-specific siRNA sequence 2; oeNC: Overexpression Empty Vector Negative Control; oeCOX7A1:COX7A1 overexpression plasmid.

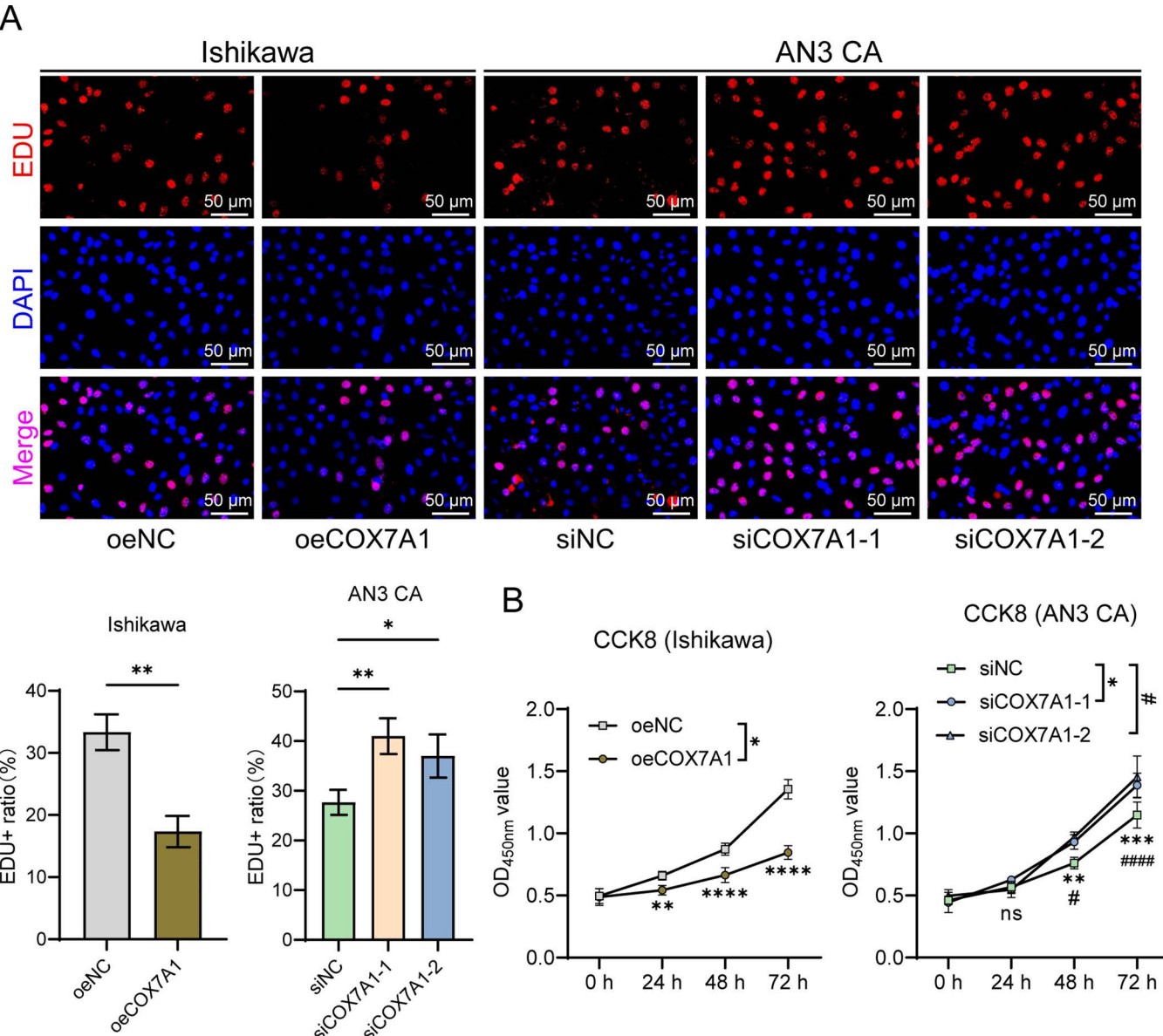

**Fig 4. COX7A1 effectively inhibits the proliferation of endometrial cancer cells.** An EDU assay was used to evaluate changes in cell proliferation ability 48 hours after transfection with the overexpression plasmid in Ishikawa cells and siRNA in AN3 CA cells (200×, Scale bar: 50 μm) (n = 3). * represents the comparisons between two groups (*$P < 0.05$, **$P < 0.01$). **(B)** Cell viability changes were assessed at 0, 24, 48, and 72 hours after transfection using the CCK-8 assay (n = 3). In Ishikawa cells, * denotes comparisons between groups, (*$P < 0.05$). In AN3 CA cells, * represents comparisons between siCOX7A1-1 and siNC (*$P < 0.05$). # signifies comparisons between siCOX7A1-2 and siNC (#$P < 0.05$). EDU: 5-Ethynyl-2'-deoxyuridine; COX7A1: Cytochrome c oxidase subunit 7A1; siNC: siRNA Negative Control; siCOX7A1-1: COX7A1-specific siRNA sequence 1; siCOX7A1-2: COX7A1-specific siRNA sequence 2; oeNC: Overexpression Empty Vector Negative Control; oeCOX7A1: COX7A1 overexpression plasmid.

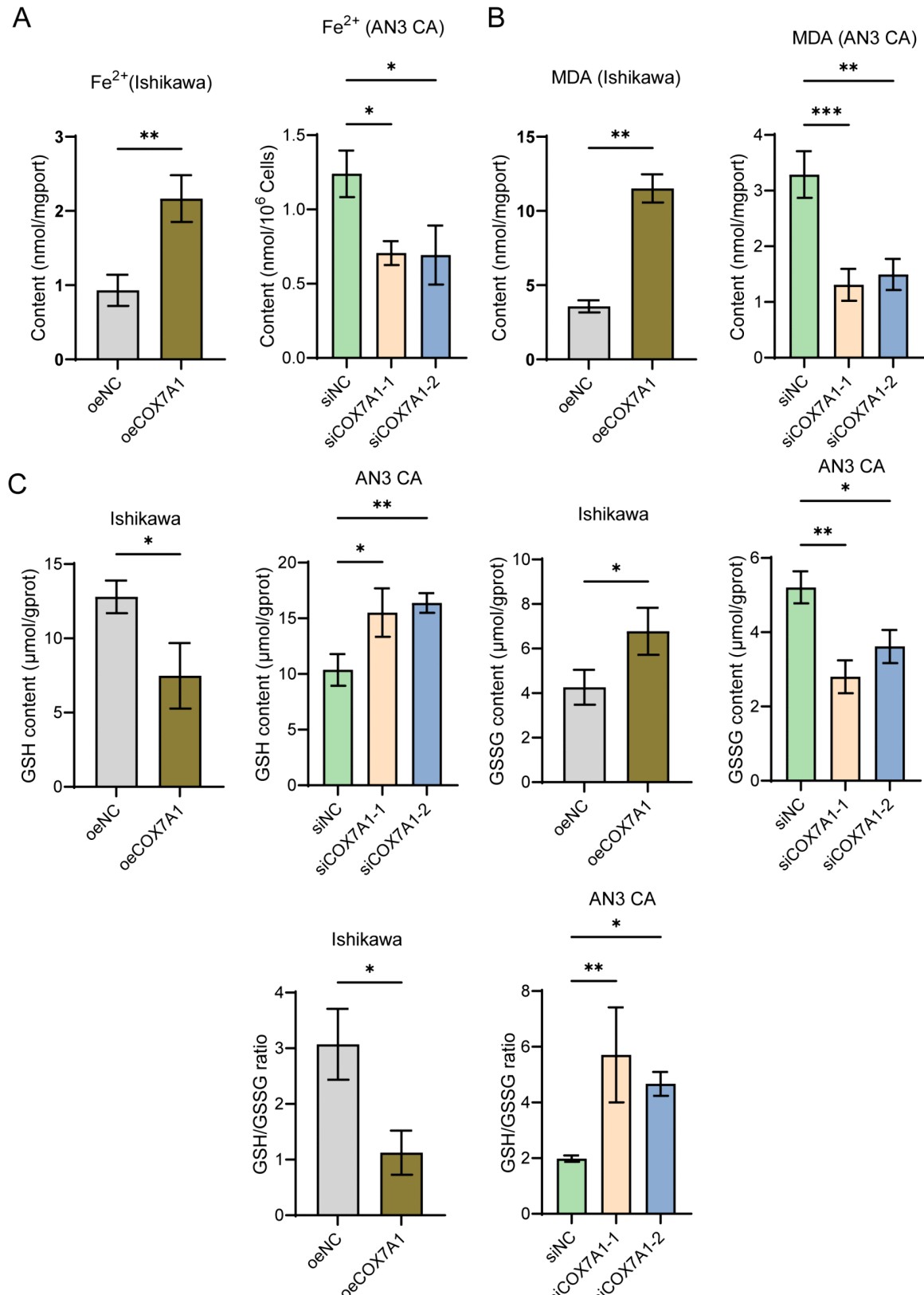

**Fig 5. COX7A1 enhances the levels of MDA and Fe$^{2+}$ and reduces the ratio of GSH/GSSG in endometrial cancer cells. (A)** After transfection with overexpression plasmid or siRNA for 48 hours in endometrial cancer cells, the changes in intracellular Fe$^{2+}$ levels were detected (n = 3). **(B)** The changes

in intracellular MDA levels were detected 48 hours after transfection in endometrial cancer cells (n = 3). **(C)** The changes in intracellular GSH and GSSG levels were detected 48 hours after transfection in endometrial cancer cells, and the changes in the GSH/GSSG ratio were statistically analyzed (n = 3). Statistical comparisons between two groups are indicated (*$P < 0.05$, **$P < 0.01$, ***$P < 0.001$). MDA: Malondialdehyde; GSH: Glutathione; GSSG: Glutathione disulfide; COX7A1: Cytochrome c oxidase subunit 7A1; siNC: siRNA Negative Control; siCOX7A1-1: COX7A1-specific siRNA sequence 1; siCOX7A1-2: COX7A1-specific siRNA sequence 2; oeNC: Overexpression Empty Vector Negative Control; oeCOX7A1: COX7A1 overexpression plasmid.

identifying key genes that are critically associated with treatment outcomes and patient prognosis remains a current research priority.

The emergence of bioinformatics analysis methods offers a robust tool to solve this problem. In this study, the ferroptosis-driving factor COX7A1 was identified through integrated bioinformatics analysis. Subsequently, in vitro experiments revealed the regulatory impact of COX7A1 expression differences on cell ferroptosis in endometrial cancer. The results of these in vitro studies showed that COX7A1 overexpression could significantly inhibit the proliferation of endometrial cancer cells, consistent with its suppressive effect observed in non-small cell lung cancer [35,36]. However, it is noteworthy that COX7A1 is highly expressed in gastric cancer and associated with poor prognosis [37], and its high expression can also enhance resistance to oxaliplatin in gastric cancer [38], suggesting that COX7A1 functions as a tumor-promoting factor in gastric cancer. This tumor-promoting role of COX7A1 in gastric cancer contrasts with reports in non-small cell lung cancer [35] and the anticancer effects observed in endometrial cancer cells in this study. We hypothesize that this discrepancy may be attributed to the complexity of the tumor microenvironment, gene regulation, cell signaling pathways, and immune microenvironment associated with COX7A1 in different cancers. This variability underscores the importance of considering cancer type-specific and pathologically distinct contexts when investigating or applying COX7A1 as a biomarker or therapeutic target. Future research should aim to uncover the specific mechanisms of COX7A1 in various cancers through multi-omics analysis and clinical validation. Regarding how COX7A1 affects the cell metabolism of endometrial cancer, our research shows that overexpression of COX7A1 leads to a significant decrease in mitochondrial membrane potential (ΔΨm) and a down-regulation of the expression of ATP5A1, a key protein in energy metabolism. This suggests that it impairs the normal energy-metabolic function of mitochondria. The overexpression of COX7A1 also causes a significant increase in intracellular reactive oxygen species (ROS) levels. Meanwhile, it is accompanied by a down-regulation of the expression of GPX4, a core antioxidant enzyme, and a decrease in the GSH/GSSG ratio, indicating that the cellular redox balance is disrupted and the antioxidant capacity is impaired. Moreover, overexpression of COX7A1 results in an increase in intracellular $Fe^{2+}$ levels and an elevation of malondialdehyde (MDA), the end-product of lipid peroxidation. At the same time, the expression of ACSL4, a protein promoting ferroptosis, is up-regulated, while the expression of SLC7A11, which is involved in cystine uptake, is down-regulated. These changes constitute the typical metabolic characteristics of ferroptosis [39]. The morphological changes such as mitochondrial shrinkage observed by electron microscopy are consistent with these findings. It should be noted that this study has verified the changes in these key metabolic nodes. However, some of the upstream-downstream molecular relationships among these nodes still need to be further clarified in future research.

In the realm of mechanistic investigation, this study elucidates that COX7A1 may regulate the ferroptosis process through multiple pathways. Specifically, it induces ferroptosis by regulating iron ion metabolism, lipid peroxidation, and mitochondrial dysfunction. Notably, overexpression of COX7A1 significantly reduced the expression levels of GPX4 and SLC7A11 while increasing the expression of ACSL4. Conversely, knockdown of COX7A1 resulted in upregulation of GPX4 and SLC7A11 and decreased ACSL4 levels. These observations are consistent with the classic ferroptosis mechanism, which induces cell death through lipid peroxidation and disruption of the antioxidant system [40]. Furthermore, COX7A1 exacerbates cellular oxidative stress and ferroptosis by disrupting mitochondrial membrane potential and enhancing outer membrane permeability. These results suggest that COX7A1 mediates mitochondrial dysfunction, thereby influencing the

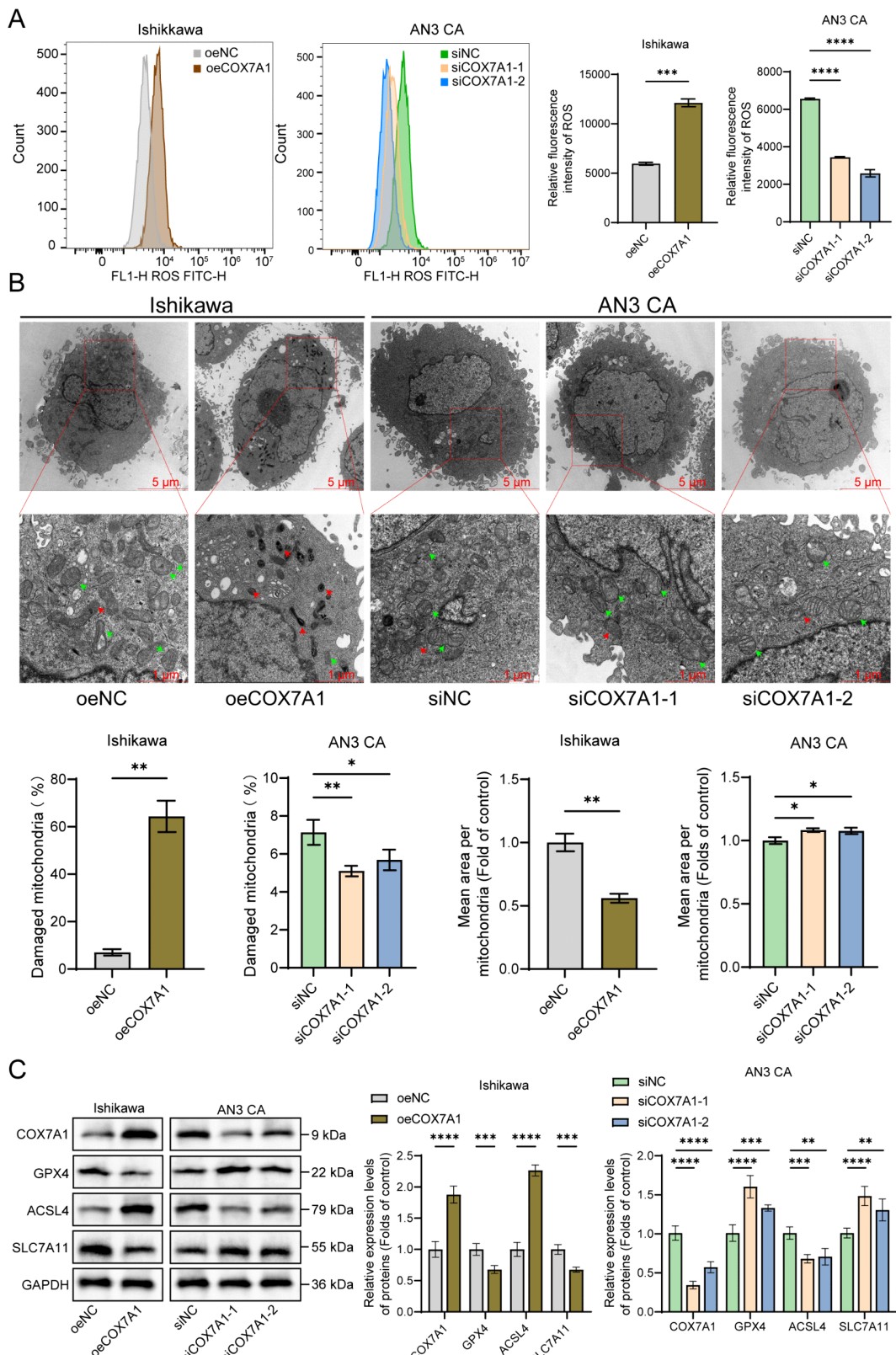

**Fig 6. COX7A1 enhances the intracellular oxidative stress level of endometrial cancer cells and induces ferroptosis.** After transfection with the overexpression plasmid or siRNA for 48 hours in endometrial cancer cells, the changes in intracellular ROS levels were detected

(n = 3). **(B)** Mitochondrial morphological changes were observed using electron microscopy. The red arrow indicates damaged mitochondria, and the green arrow indicates normal mitochondria (n = 3). (The first column of images: 2000×, Scale bar: 5 µm. The second column of images: 5000×, Scale bar: 1 µm). **(C)** The relative expression levels of COX7A1, GPX4, ACSL4, and SLC7A11 proteins in endometrial cancer cells were detected by Western blot (n = 3). Statistical comparisons between two groups are indicated (**$P < 0.01$, ***$P < 0.001$, ****$P < 0.0001$). JC-1: 5,5',6,6'-Tetrachloro-1,1',3,3'-tetraethylbenzimidazolylcarbocyanine iodide; VDAC1: Voltage-dependent anion-selective channel 1; Cytochrome C: Cytochrome c; ATP5A1: ATP synthase F1 subunit alpha; OPA1: Optic atrophy 1; COX7A1: Cytochrome c oxidase subunit 7A1; siNC: siRNA Negative Control; siCOX7A1-1: COX7A1-specific siRNA sequence 1; siCOX7A1-2: COX7A1-specific siRNA sequence 2; oeNC: Overexpression Empty Vector Negative Control; oeCOX7A1: COX7A1 overexpression plasmid.

ferroptosis process. It is important to note that this represents our hypothesis regarding the mechanism by which COX7A1 induces ferroptosis in endometrial cancer, and further experimental validation is required. Consistent with our findings, Feng Y et al. reported that COX7A1 enhances the sensitivity of human NSCLC cells to cystine deprivation-induced ferroptosis by regulating mitochondrial metabolism [41]. This finding corroborates our observation that COX7A1 can potentiate ferroptosis sensitivity, thereby reinforcing the reliability of our conclusions. Research has demonstrated that, in addition to the classical GPX4 pathway, phospholipid hydroperoxides serve as executors of ferroptosis and are associated with GPX4-independent monitoring pathways [42,43]. While current evidence suggests that COX7A1 may regulate ferroptosis through the GPX4 pathway, its role in other ferroptosis-inducing mechanisms remains to be elucidated. This represents a key area for future investigation.

In conclusion, this study systematically explored the expression profile of COX7A1 in endometrial cancer and its regulatory impact on ferroptosis for the first time, filling a significant gap in existing research. By revealing the crucial role of COX7A1 in ferroptosis, this study not only provides a new perspective for understanding the molecular mechanism underlying endometrial cancer but also identifies a potential therapeutic target for developing innovative ferroptosis-based treatment strategies. Despite the promising findings, several limitations warrant acknowledgment. First, this study primarily relies on in vitro cell experiments, which lack verification from in vivo models, thus limiting our understanding of COX7A1's role within the tumor microenvironment. Secondly, although the relationship between COX7A1 and ferroptosis has been explored, the precise mechanisms governing its influence on cellular metabolism and signaling pathways remain to be fully elucidated. Finally, there are several deficiencies in methodological validation. For instance, in the EdU assay, we performed a semi-quantitative analysis without verifying the number of EdU-positive cells via flow cytometry. In the CCK-8 assay, we determined cell viability solely by measuring optical density (OD) values, without employing direct cell counting to quantify the exact cell numbers. Furthermore, when constructing the knockdown and overexpression cell lines, we did not perform bidirectional verification (i.e., both gain-of-function and loss-of-function experiments) across all cell lines used. There are limitations in the detection dimensions of lipid peroxidation products. We have listed "systematically analyzing different lipid peroxidation intermediates and end-products using techniques such as lipidomics" as an important future research direction. Future research should aim to explore the function of COX7A1 in in vivo models and integrate multi-omics techniques to comprehensively analyze its molecular mechanisms in regulating ferroptosis. Moreover, given the interplay between ferroptosis and other forms of cell death (such as apoptosis and necrosis), future studies could explore the synergistic effects of COX7A1 in multiple cell death pathways.

## Conclusion

This study demonstrates that COX7A1 is a key regulator of ferroptosis in endometrial cancer cells. Overexpression of COX7A1 drives the ferroptosis process by disrupting intracellular iron homeostasis, inducing oxidative stress, and impairing mitochondrial function. In contrast, low expression of COX7A1 exhibits the opposite effect, promoting tumor cell proliferation. These findings not only reveal a novel mechanism by which COX7A1 inhibits the progression of endometrial cancer through the ferroptosis pathway but also suggest its potential as a therapeutic target for ferroptosis-based therapies. This provides new directions and a theoretical foundation for the treatment strategies of endometrial cancer.

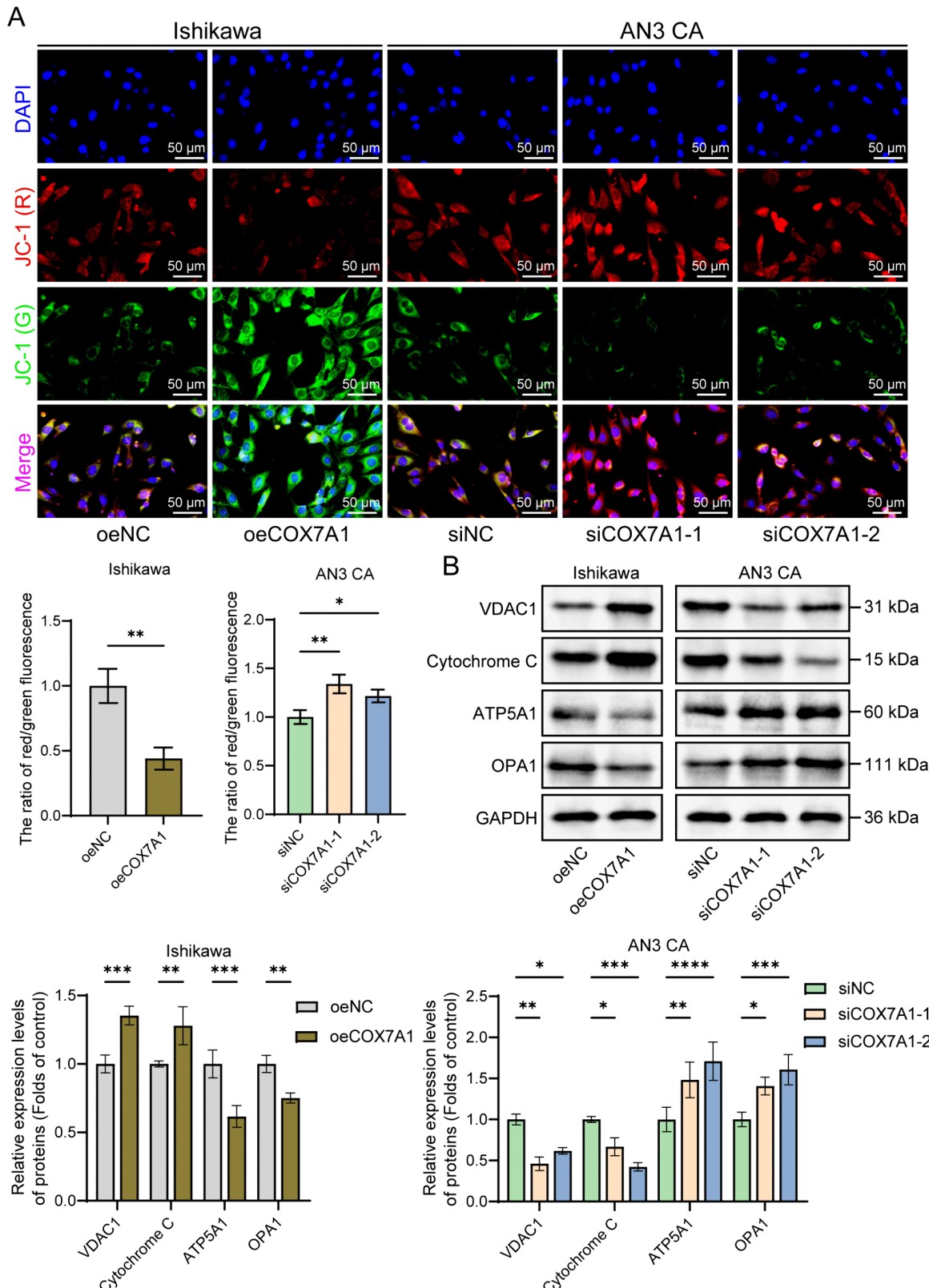

**Fig 7. COX7A1 disrupts the mitochondrial membrane potential and induces mitochondrial dysfunction. (A)** JC-1 was used to detect changes in mitochondrial membrane potential in endometrial cancer cells after different treatments (200×, Scale bar: 50 μm) (n = 3). **(B)** The relative expression

levels of VDAC1, Cytochrome C, ATP5A1, and OPA1 proteins in endometrial cancer cells were detected by Western blot analysis (n = 3). Statistical comparisons between two groups are indicated (*$P$ < 0.05, **$P$ < 0.01, ***$P$ < 0.001, ****$P$ < 0.0001). ROS: Reactive oxygen species; COX7A1: Cytochrome c oxidase subunit 7A1; GPX4: Glutathione peroxidase 4; ACSL4: Acyl-CoA synthetase long-chain family member 4; SLC7A11: Solute carrier family 7 member 11; siNC: siRNA Negative Control; siCOX7A1-1: COX7A1-specific siRNA sequence 1; siCOX7A1-2: COX7A1-specific siRNA sequence 2; oeNC: Overexpression Empty Vector Negative Control; oeCOX7A1:COX7A1 overexpression plasmid.

## Supporting information

**S1 File. Raw images.**
(PDF)

**S2 File. Raw material.**
(PDF)

**S1 Table. RT-qPCR primer sequence, COX7A1 interference sequence and COX7A1 overexpression sequence.**
(XLSX)

**S2 Table. Information and dilution ratios of antibodies for Western blot experiments.**
(XLSX)

## Author contributions

**Conceptualization:** Qi Wu, Lina Han, Pei Wang.

**Data curation:** Qi Wu.

**Formal analysis:** Lina Han, Pei Wang.

**Investigation:** Qi Wu, Liyun Song.

**Methodology:** Qi Wu.

**Project administration:** Pei Wang.

**Resources:** Suning Bai, Pei Wang.

**Software:** Suning Bai.

**Supervision:** Pei Wang.

**Validation:** Qi Wu, Suning Bai, Liyun Song.

**Visualization:** Qi Wu.

**Writing – original draft:** Qi Wu, Lina Han.

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
