## [Decision Letter · Decision Letter 0]

29 Oct 2025

Dear Dr. Wu,

Thank you for submitting your manuscript to PLOS ONE. After careful consideration, we feel that it has merit but does not fully meet PLOS ONE’s publication criteria as it currently stands. Therefore, we invite you to submit a revised version of the manuscript that addresses the points raised during the review process.

We look forward to receiving your revised manuscript.

Kind regards,

Mahesh Narayan, Ph.D.

Academic Editor

PLOS ONE

Journal Requirements:

5. In the online submission form, you indicated that the dataset associated with this study can be obtained from the lead researcher if a justified request is made.

Reviewers' comments:

Reviewer's Responses to Questions

**Comments to the Author**

1. Is the manuscript technically sound, and do the data support the conclusions?

Reviewer #1: Partly

Reviewer #2: Yes

2. Has the statistical analysis been performed appropriately and rigorously?

Reviewer #1: Yes

Reviewer #2: Yes

3. Have the authors made all data underlying the findings in their manuscript fully available?

Reviewer #1: Yes

Reviewer #2: Yes

4. Is the manuscript presented in an intelligible fashion and written in standard English?

Reviewer #1: Yes

Reviewer #2: Yes

Reviewer #1: COX7A1-mediated mitochondrial dysfunction can induce ferroptosis in the endometrial

cancer cells

The study investigates the role of COX7A1 in mitochondrial dysfunction and ferroptosis in endometrial cancer. Bioinformatics analysis identified differentially expressed genes related to ferroptosis, and in vitro experiments confirmed that COX7A1 overexpression inhibits cancer cell proliferation and induces ferroptosis cell death. COX7A1 increased intracellular Fe2+ and MDA levels, reduced the GSH/GSSG ratio, and altered mitochondrial membrane potential. The findings of this study imply the significance of the COX7A1 gene as a driver of mitochondrial dysfunction and cellular stress in endometrial cancer cells. Although this gene has been investigated in other cancers, studying its role in endometrial cancer is also important. However, I have the following comments and concerns regarding the manuscript.

1) The abstract, conclusion, and discussion primarily focus on overexpression data obtained from Ishikawa cells. There should be a discussion on knockdown in AN3 CA cells and why this marker was highly expressed in these cells. We observed that almost all endometrial cancer cell lines expressed COX7A1, except for Ishikawa cells, suggesting a potential regulatory role of this gene in these cancers, which warrants further discussion. The results based on Ishikawa cells cannot be generalized to other pancreatic cancer cells unless data are available showing the same result of the overexpression in the cells, which already expresses COX7A1.

2) The absence of COX7A1 in Ishikawa cells suggests that the absence of this gene contributes to the invasive nature of these cells. In contrast, why is the COX7A1 gene highly expressed in AN3 CA cells? Does this indicate that these cancer cells are less invasive and aggressive?

3) Figure 6A: What is the quantified percent change in ROS? Please mention the percentage count of cells that expressed ROS.

4) Figure 3 B: Data after transfection, the relative change was measured. Please add transfection efficiency data in the supplemental file. Additionally, please include the protein level expression of COX7A1 after transfection.

5) What is the difference btw siCOx7A1-1 and siCOx7A1-2? What is your OENC and SiNC? Did you select (transfected cell selection) the knockdown and overexpression cells and then use them for the downstream experiments?

6) For your cck8 experiment, did you normalize your data against cell number and background absorbance? What does your OD data tell you about the cell state? The reagent (tetrazolium salt) provides information about the metabolic state of cells, but did you perform cell counting to assess cell proliferation? If so, please mention the data.

7) After plating cells for EDU, how long did you incubate before adding the EDU solution? The fluorescence images just represent a selected part of the well. How did you quantify and count edu positive cells? Did you take images of all sites of the well, or did you perform flow cytometry to count the cells?

8) Why did you choose to do the knockdown for COX7A1 in AN3 CA rather than overexpressing it? What purpose does the presence of COX7A1 in these cells serve? If we overexpress COX7A1 in AN3 CA rather than knock it down, will we experience the same inhibitory effects?

Minor comments :

1) Explain the abbreviations in figure legends like NC, siNC, oeNC, and all the markers you studied.

2) Please improve the grammar of the abstract, e.g, Line 2 of the abstract should say "we explored.." and also line 8 of the abstract should read: 'Effect of COX7A1 on the inhibition of endometrial cancer cell growth.'

3) In the legends, please mention the number of replicates or experiments for each figure.

4) The ferroptosis and cell stress induction mechanism of COX7A1 can be observed in other cancers as well, such as lung and gastric cancers. Make sure you add relevant references in the manuscript. ( https://www.nature.com/articles/s41419-022-05430-3, https://www.sciencedirect.com/science/article/pii/S0006291X16319659,
https://eurjmedres.biomedcentral.com/articles/10.1186

Reviewer #2: Comments to authors (s)

The research article presented by Qi Wu et al. on COX7A1-mediated mitochondrial dysfunction can induce ferroptosis in endometrial cancer cells has a minor issue with the manuscript.

The authors need to address the following comments.

•Please elaborate the material and methods section, like the double antibiotics name, trypsin solution dose, and full abbreviation of buffer, etc., for better understanding.

•Do the authors think that the COX7RP gene is also associated with ferroptosis in endometrial cancer, like COX7A1?

•How does the COX7A1 affect the cell metabolism in endometrial cancer?

•Why do the authors think that the microenvironment of different cancers can regulate the COX7A1 and how?

•Do the authors think monitoring other lipid peroxidation products would also be beneficial?

**Do you want your identity to be public for this peer review?** For information about this choice, including consent withdrawal, please see our Privacy Policy

Reviewer #1: **Yes:** Sonia Kiran

Reviewer #2: **Yes:** ABHIRUCHI KANT

---

## [Author Response · Author response to Decision Letter 1]

12 Jan 2026

Journal Requirements:

Response: We have edited our manuscript strictly in accordance with the format requirements to ensure it meets the requirements of your journal.

Response: We have uploaded all the original R codes involved in the bioinformatics research of this study to the online system for inspection (https://figshare.com/s/5678c40e643bf55f363e).

Response: We have uploaded all the original Western blot bands to the online database (https://figshare.com/s/5678c40e643bf55f363e).

4.We note that you have indicated that there are restrictions to data sharing for this study. PLOS only allows data to be available upon request if there are legal or ethical restrictions on sharing data publicly. For more information on unacceptable data access restrictions, please see http://journals.plos.org/plosone/s/data-availability#loc-unacceptable-data-access-restrictions.

In the online submission form, you indicated that the dataset associated with this study can be obtained from the lead researcher if a justified request is made.

All PLOS journals now require all data underlying the findings described in their manuscript to be freely available to other researchers, either 1. In a public repository, Within the manuscript itself, or 3. Uploaded as supplementary information.

Response: We have updated the data availability statement and stored the original data of this study on the online website (https://figshare.com/s/5678c40e643bf55f363e).

Response: We have added the "Supporting Information" at the end of the article.

Response: We have read the literatures recommended by the reviewers, which are highly relevant to our research. Therefore, we have cited these literatures as required by the reviewers. Among them, one has been cited before, one is newly added, and for another one, due to incomplete information provided by the reviewers, we haven't been able to locate the corresponding literature.

Reviewers' comments:

Reviewer's Responses to Questions

Comments to the Author

1. Is the manuscript technically sound, and do the data support the conclusions?

Reviewer #1: Partly

Reviewer #2: Yes

2. Has the statistical analysis been performed appropriately and rigorously?

Reviewer #1: Yes

Reviewer #2: Yes

3. Have the authors made all data underlying the findings in their manuscript fully available?

Reviewer #1: Yes

Reviewer #2: Yes

4. Is the manuscript presented in an intelligible fashion and written in standard English?

Reviewer #1: Yes

Reviewer #2: Yes

5. Review Comments to the Author

Reviewer #1: COX7A1-mediated mitochondrial dysfunction can induce ferroptosis in the endometrial

cancer cells

The study investigates the role of COX7A1 in mitochondrial dysfunction and ferroptosis in endometrial cancer. Bioinformatics analysis identified differentially expressed genes related to ferroptosis, and in vitro experiments confirmed that COX7A1 overexpression inhibits cancer cell proliferation and induces ferroptosis cell death. COX7A1 increased intracellular Fe2+ and MDA levels, reduced the GSH/GSSG ratio, and altered mitochondrial membrane potential. The findings of this study imply the significance of the COX7A1 gene as a driver of mitochondrial dysfunction and cellular stress in endometrial cancer cells. Although this gene has been investigated in other cancers, studying its role in endometrial cancer is also important. However, I have the following comments and concerns regarding the manuscript.

1) The abstract, conclusion, and discussion primarily focus on overexpression data obtained from Ishikawa cells. There should be a discussion on knockdown in AN3 CA cells and why this marker was highly expressed in these cells.

Response: Thank you for your valuable comments. We agree with your perspective. We have added the relevant changes in proteins and ferroptosis after AN3 CA knockdown in the abstract, conclusion, and discussion sections.

Main revisions in the abstract:

In vitro experiments have demonstrated that overexpression of COX7A1 inhibits the proliferation of endometrial cancer cells and induces ferroptosis by regulating intracellular iron metabolism and mitochondrial function. The specific mechanisms include increasing intracellular Fe²⁺ and malondialdehyde (MDA) levels, decreasing the GSH/GSSG ratio, and disrupting mitochondrial membrane potential, thereby leading to mitochondrial dysfunction. Furthermore, COX7A1 overexpression significantly reduces the expression of glutathione peroxidase 4 (GPX4) and SLC7A11, while upregulating acyl-coenzyme A synthetase long-chain family member 4 (ACSL4). In contrast, knockdown of COX7A1 promotes the proliferation of endometrial cancer cells and inhibits ferroptosis, exhibiting the opposite effects. These findings provide new insights into the molecular mechanisms of endometrial cancer Lines 39 - 49�.

Main revisions in the discussion:

Notably, overexpression of COX7A1 significantly reduced the expression levels of GPX4 and SLC7A11 while increasing the expression of ACSL4. Conversely, knockdown of COX7A1 resulted in upregulation of GPX4 and SLC7A11 and decreased ACSL4 levels. These observations are consistent with the classic ferroptosis mechanism, which induces cell death through lipid peroxidation and disruption of the antioxidant system �Lines 434-439�.

Main revisions in the conclusion:

In contrast, low expression of COX7A1 exhibits the opposite effect, promoting tumor cell proliferation. These findings not only reveal a novel mechanism by which COX7A1 inhibits the progression of endometrial cancer through the ferroptosis pathway but also suggest its potential as a therapeutic target for ferroptosis-based therapies Lines 481-486�.

We observed that almost all endometrial cancer cell lines expressed COX7A1, except for Ishikawa cells, suggesting a potential regulatory role of this gene in these cancers, which warrants further discussion. The results based on Ishikawa cells cannot be generalized to other pancreatic cancer cells unless data are available showing the same result of the overexpression in the cells, which already expresses COX7A1.

Response: We completely agree with your opinion. To elaborate our viewpoints more clearly and respond to your questions, we are providing more detailed background and explanations here, hoping to dispel your concerns:

One of the core and consistent findings in our study is that the basal expression of COX7A1 can be detected in the vast majority of endometrial cancer cell lines we examined. Significantly, compared with normal cells (HEEpiC - SV40), the expression of COX7A1 is lower in endometrial cancer cell lines, and this difference is statistically significant (Figure 3A).

Our central hypothesis is that COX7A1 is a tumor suppressor with down - regulated expression in cancer, and the inactivation or reduced expression of its function is conducive to tumor cells evading ferroptosis.

Therefore, in Ishikawa cells, we demonstrated that "introducing" COX7A1 can inhibit tumor growth.

Conversely, knocking down the basal - expressed COX7A1 in these cells promotes tumor proliferation (as described in the original text).

These two sets of experimental evidence, one positive and one negative, form a complete logical loop under multiple cell backgrounds, jointly supporting the conclusion that COX7A1 acts as a broad - spectrum regulatory factor. Its biological effects do not depend on whether a specific cell line expresses it "from scratch", but rather on its relative expression level and functional state.

2) The absence of COX7A1 in Ishikawa cells suggests that the absence of this gene contributes to the invasive nature of these cells. In contrast, why is the COX7A1 gene highly expressed in AN3 CA cells? Does this indicate that these cancer cells are less invasive and aggressive?

Response: Thank you for raising such profound questions. You've astutely observed a key phenomenon in our data: COX7A1 is absent in Ishikawa cells but highly expressed in AN3 CA cells, and you've logically questioned its impact on cell invasiveness. This is indeed a point worthy of in - depth exploration.

First of all, we fully agree with your observation. The differences in COX7A1 expression levels among different cell lines are likely to reflect the heterogeneity of endometrial cancer and its different molecular subtypes. Our core explanation for this phenomenon is that the absolute expression level of a single gene is not always in a simple linear negative correlation with the final cell phenotype (such as invasiveness).

Despite the differences in absolute levels, our research has revealed a more crucial common pattern:

Compared with normal tissues: As you know, we have clarified in the paper that in endometrial cancer tissue samples, the expression of COX7A1 is generally significantly lower than that in paired normal endometrial tissues. Even though the basal expression level of COX7A1 in AN3 CA cells is relatively high, our functional experimental data show that further knocking down COX7A1 in these cells can still significantly promote cell proliferation (Figure 4A). Conversely, overexpressing COX7A1 in Ishikawa cells can also lead to a further inhibition of cell proliferation.

However, the main focus of this study was on discussing the effects of COX7A1 on the proliferation, ferroptosis, and mitochondrial dysfunction of endometrial cancer. We acknowledge that the basal expression of COX7A1 may have varying degrees of impact on the invasion and migration of different endometrial cancer cells, but this still requires further investigation in our subsequent experiments. Once again, thank you for your valuable comments, which have given us a deeper understanding and insights into the research direction and cell line selection of this project.

3) Figure 6A: What is the quantified percent change in ROS? Please mention the percentage count of cells that expressed ROS.

Response: Thank you for raising this crucial question. We have re - composed the figures in terms of percentages as follows. We've noticed that in most papers, presenting the average fluorescence intensity of ROS and the positive ratio as output results are both quite common. If you need us to change it to the ROS positive ratio, please let us know. We will be happy to make the modification.

4) Figure 3 B: Data after transfection, the relative change was measured. Please add transfection efficiency data in the supplemental file. Additionally, please include the protein level expression of COX7A1 after transf

---

## [Editor Report · Decision Letter 1]

22 Jan 2026

COX7A1-mediated mitochondrial dysfunction can induce ferroptosis in endometrial cancer cells

PONE-D-25-43159R1

Dear Dr. Wu,

We’re pleased to inform you that your manuscript has been judged scientifically suitable for publication and will be formally accepted for publication once it meets all outstanding technical requirements.

Kind regards,

Mahesh Narayan, Ph.D.

Academic Editor

PLOS One

Additional Editor Comments (optional):

This is satisfactory
---

## [Editor Report · Acceptance letter]

PONE-D-25-43159R1

PLOS One

Dear Dr. Wu,

I'm pleased to inform you that your manuscript has been deemed suitable for publication in PLOS One. Congratulations! Your manuscript is now being handed over to our production team.

Kind regards,

on behalf of

Dr. Mahesh Narayan

Academic Editor

PLOS One